# A Human Cellular Model for Colorectal Anastomotic Repair: The Effect of Localization and Transforming Growth Factor-β1 Treatment on Collagen Deposition and Biomarkers

**DOI:** 10.3390/ijms22041616

**Published:** 2021-02-05

**Authors:** Ceylan Türlü, Nicholas Willumsen, Debora Marando, Peter Schjerling, Edyta Biskup, Jens Hannibal, Lars N. Jorgensen, Magnus S. Ågren

**Affiliations:** 1Digestive Disease Center, Bispebjerg Hospital, University of Copenhagen, 2400 Copenhagen, Denmark; ceylanturlu@outlook.dk (C.T.); deboramarando94@gmail.com (D.M.); larsnjorgensen@hotmail.com (L.N.J.); 2Nordic Bioscience A/S, 2730 Herlev, Denmark; nwi@NordicBio.com; 3Institute of Sports Medicine Copenhagen, Department of Orthopedic Surgery, Copenhagen University Hospital—Bispebjerg and Frederiksberg, 2400 Copenhagen, Denmark; Peter@mRNA.dk; 4Center for Healthy Aging, Department of Clinical Medicine, University of Copenhagen, 2200 Copenhagen, Denmark; 5Department of Dermatology and Copenhagen Wound Healing Center, Bispebjerg Hospital, University of Copenhagen, 2400 Copenhagen, Denmark; edyta.biskup@gmail.com; 6Department of Clinical Biochemistry, Bispebjerg Hospital, University of Copenhagen, 2400 Copenhagen, Denmark; Jens.Hannibal@regionh.dk; 7Department of Clinical Medicine, Faculty of Health and Medical Sciences, University of Copenhagen, 2200 Copenhagen, Denmark

**Keywords:** extracellular matrix, collagen, wound healing, growth factors, colon, rectum, anastomotic leakage

## Abstract

Anastomotic leakage (AL) is a devastating complication after colorectal surgery, possibly due to the loss of stabilizing collagen fibers in the submucosa. Our aim was to assess the formation of collagen in the colon versus the rectum with or without transforming growth factor (TGF)-β1 exposure in a human cellular model of colorectal repair. Primary fibroblasts were isolated by an explant procedure from clinically resected tissue rings during anastomosis construction in 19 consecutive colorectal patients who underwent laparoscopy. The cells, identified as fibroblasts by morphologic characteristics and flow cytometry analysis (CD90^+^), were cultured for 8 days and in 12 patients in the presence of 1 ng/mL TGF-β1. Total collagen deposition was measured colorimetrically after Sirius red staining of fixed cell layers, and type I, III, and VI collagen biosynthesis and degradation were specifically determined by the biomarkers PINP, PRO-C3, PRO-C6, and C3M in conditioned media by competitive enzyme-linked immunosorbent assays. Total collagen deposition by fibroblasts from the colon and rectum did not significantly differ. TGF-β1 treatment increased PINP, PRO-C6, and total collagen deposition. Mechanistically, TGF-β1 treatment increased *COL1A1* and *ACTA2* (encoding α-smooth muscle actin), and decreased *COL6A1* and *MMP2* mRNA levels in colorectal fibroblasts. In conclusion, we found no effect of anatomic localization on collagen production by fibroblasts derived from the large intestine. TGF-β1 represents a potential therapeutic agent for the prevention of AL by increasing type I collagen synthesis and collagen deposition.

## 1. Introduction

Anastomotic leakage (AL) is the most feared complication after colorectal surgery [1,2,3]. AL is associated with increased morbidity, cancer recurrence, and mortality [2]. The incidence is 3–6% after colonic resection and 10–20% after rectal resection [1,2,3,4]. There is an unmet need for effective pharmaceuticals to prevent AL [5,6,7]. Apart from experimental animal models [5], there is a paucity of human models for delineation of pathophysiological mechanisms and screening novel interventions [8]. The use of a human model for anastomotic wound repair would be extremely valuable before embarking on resource-demanding randomized controlled trials [9].

The extracellular matrix (ECM) is key for maintaining homeostasis of the large bowel [10,11]. In particular, the interstitial collagens in the submucosa are paramount for the strength of the colorectum [12,13]. Type I and III collagens are the most abundant collagens of the bowel wall [14]. Tissue resection during surgery initiates a wound repair response [15]. In the immediate postoperative period, a dramatic reduction of collagens occurs due to the action of matrix metalloproteinases (MMPs) [14,16]. Broad-spectrum MMP inhibitors are also effective in preclinical models of normal anastomotic wound healing but these compounds also have harmful effects [17]. The synthesis of new collagen molecules, predominantly fibrillar type I and type III collagens, begins later and peaks approximately 1 week postoperatively [18,19]. Nonfibrillar type VI collagen encircles the type I and III collagen fibers, promoting fibrillogenesis, and may contribute to the quality of the new interstitial collagens [20,21,22,23,24]. Thus, any anabolic therapeutic with the capacity to compensate for the loss of collagen early in the wound healing cascade is an attractive intervention [5,6].

Fibroblasts are key cells in the production of type I, III, and VI collagens [25]. Growth factors, and transforming growth factor (TGF)-β1 in particular, are important mediators of ECM formation and wound healing [5,15,25,26,27,28,29,30,31]. Notably, increased TGF-β1 expression coincides with *COL1A1* mRNA levels in an experimental model of anastomotic wound healing [32]. In another study, TGF-β1 protein levels increased in anastomosed colon and were correlated with soluble collagen in colon anastomoses, indicating a direct connection between TGF-β1 and collagen biosynthesis during anastomotic wound healing [33]. Finally, an interventional study with TGF-β1 gene transfer indicated beneficial effects on anastomotic repair [34]. Thus, TGF-β1 is a potential therapeutic agent due to its profound effect on collagen synthesis and stimulation of anastomotic wound healing, possibly leading to reduced rates of AL.

We recently demonstrated decreased *COL1A1* and *COL3A1* mRNA levels in the rectum compared to colon, which may explain the higher risk of AL after rectal resection compared to colonic resection [35]. The primary aim of this study (DONUT2) was to (1) compare collagen production using fibroblasts isolated from the colon and rectum and (2) determine the effect of TGF-β1 on isolated fibroblasts. Total collagen deposition was measured with Sirius red staining [29], and specific type I, type III, and type VI collagen biosynthesis was quantified by enzyme-linked immunosorbent assays (ELISAs) with antibodies directed against epitopes of the propeptides of type I collagen (PINP), type III collagen (PRO-C3), and type VI collagen (PRO-C6) released by the fibroblasts into the conditioned medium [36,37,38]. MMP activity was measured using the biomarker C3M [39].

## 2. Results

Thirty-three eligible patients scheduled for laparoscopic sigmoid resection (SR) or low anterior resection (LAR) were approached for participation the day before their operations from 2 October 2017 to 18 January 2018, and 31 patients were included after providing their written consent. Tissue samples from 10 patients were excluded for various reasons as indicated in Figure 1. From the remaining 21 patients, fibroblasts were isolated from the proximal and distal tissue rings, representing the colon and rectum, respectively, procured during surgery; successful outgrowth was achieved in explants from 19 patients, as shown in Figure 1. Three of the 31 included patients and one of the 19 analyzed patients developed AL within the 1-month observation period.

### 2.1. Characterization of Isolated Cells

We applied an explant procedure to obtain primary fibroblasts [40]. This method has been used previously to isolate colonic and rectal fibroblasts [31,41,42].

Cells had the typical spindle-shaped appearance of fibroblasts, as shown in Figure 2A. The cell surface antigen CD90, which is expressed in various fibroblasts [43,44,45], was analyzed by flow cytometry using a specific antibody. Fibroblasts from proximal and distal tissue rings obtained from six patients were analyzed. These analyses showed a clear separation between cells incubated with the CD90 antibody and cells incubated with the isotype control, or phosphate-buffered saline (PBS), as shown in Figure 2B.

### 2.2. Effect of Anatomic Localization on Collagen Deposition and Collagen Biomarkers

Fibroblasts of the third passage at a mean of 45 days (range: 33–72 days) from the harvest of the tissues were used for assays in 6-well tissue culture plates [29]. The amount of collagen deposited by fibroblasts from proximal and distal tissue segments did not differ significantly, as shown in Figure 3.

Furthermore, we observed no difference in fibroblast density (*p* = 0.361), measured by crystal violet-stained fixed cells, derived from proximal (OD_590nm_: 0.90 ± 0.18/well) or distal tissue rings (OD_590nm_: 0.70 ± 0.072/well).

To evaluate the turnover of different major fibroblast-derived collagens in this model system, we measured specific fragments of type I (PINP), type III (PRO-C3 and C3M), and type VI (PRO-C6) collagens released into the culture medium. The lack of difference in the amount of deposited insoluble collagen between the proximal and distal locations agrees with the collagen biomarkers released into the conditioned media, except for accumulated PRO-C6, which was higher in rectal versus colonic fibroblasts, as shown in Table 1.

### 2.3. Effect of Sex on Collagen Deposition and Biomarkers

AL occurs more often in male versus female patients [33]. Therefore, we compared the results between males and females. Because no significant difference was found between fibroblasts from the proximal and distal tissues in collagen deposition or cell growth, we pooled the data obtained from the proximal and distal fibroblasts of each patient. For this reason, the cells will henceforth be referred to as colorectal fibroblasts (CoReFs).

Collagen deposition did not differ (*p* = 0.244) between CoReFs isolated from male (OD_540nm_: 0.18 ± 0.012/well) and female (OD_540nm_: 0.20 ± 0.020/well) patients. The corresponding cell densities of the CoReFs were OD_590nm_: 0.78 ± 0.10/well for males and OD_590nm_: 0.82 ± 0.12/well for females (*p* = 0.784).

None of the collagen biomarkers in the conditioned media differed significantly between the male and female subgroups, as shown in Table 2.

### 2.4. Effect of TGF-β1 on Collagen Deposition and Cell Density

The effect of TGF-β1 was examined in fibroblast strains isolated from the last 12 analyzed patients. TGF-β1 treatment for 8 days of CoReFs increased collagen deposition, as shown in Figure 4.

Concomitantly, we observed significantly increased numbers of CoReFs treated with TGF-β1 (OD_590nm_: 1.09 ± 0.12/well, *n* = 12) (*p* < 0.0001) compared to the control (OD_590nm_: 0.76 ± 0.097/well, *n* = 12). Representative staining results are shown in Figure 5. We observed a prominent difference after trypsin treatment between control and TGF-β1-treated cultures; control CoReFs typically dissociated as single cells, whereas TGF-β1-treated CoReFs became loose, as a cohesive sheet.

### 2.5. Effect of TGF-β1 on Collagen Synthesis and Degradation Biomarkers in Conditioned Media

The cumulative PINP and PRO-C6 released in the medium over 8 days was higher in the presence of TGF-β1, while the total amount of PRO-C3 released did not differ between cells treated with TGF-β1 and control-treated cells. The released C3M in conditioned media did not differ between the control and TGF-β1 groups, as shown in Table 3.

### 2.6. Effect of TGF-β1 on COL1A1, COL3A1, COL6A1, MMP2, and ACTA2 mRNA Levels, DNA Synthesis, and α-SMA Protein Expression

To elucidate the mode of action of TGF-β1, we measured levels of key genes by RT-qPCR. TGF-β1 treatment increased *COL1A1* (1.4-fold) and *ACTA2* (8.3-fold) mRNA levels, while the normalized *COL6A1* and *MMP2* mRNA levels decreased in response to TGF-β1 treatment, as shown in Figure 6A. TGF-β1 treatment also increased the formation of formazan from cleaved water-soluble tetrazolium salt (WST-1) as a proxy for cell numbers but decreased BrdU incorporation, reflecting DNA synthesis in the CoReFs, as shown in Figure 6B. No apparent differences in morphologic features were found on examination by phase contrast microscopy, as shown in Figure 6C,D.

Immunofluorescence analyses of CoReFs indicated a low level of basal α-smooth muscle actin (α-SMA) expression, which increased considerably in response to TGF-β1, as shown in Figure 7.

### 2.7. Dose-Dependent Effect of TGF-β1 on Collagen Deposition, Cell Growth, and Release of MMP-2 into Conditioned Media

The dose-dependency of TGF-β1 on collagen deposition, cell density, and MMP-2 was studied in six different CoReF strains.

TGF-β1 treatment increased collagen deposition and cell growth in a dose-dependent manner, as shown in Figure 8A,B. There were no significant effects (*p* = 0.797) of TGF-β1 on MMP-2 levels in the conditioned media on Day 8. Results from three of the six tested CoReF strains are shown in Figure 8C. Estimated MMP-2 levels in conditioned medium on Day 8 of control-treated CoReFs were 30 ± 2.4 ng/mL compared to 29 ± 1.9 ng/mL for TGF-β1 (1 ng/mL)-treated CoReFs.

## 3. Discussion

AL is a serious complication following colorectal surgery and has been attributed to altered ECM metabolism, especially collagen metabolism [14]. In the present study, we set out to mimic the wound repair process by isolating fibroblasts from the intestinal tissue that are crucial for anastomotic wound healing [18,19,27]. Cells from colonic and rectal tissues express the fibroblast surface marker CD90 [44,46]. Furthermore, induction of *ACTA2* (α-SMA) with TGF-β1 presupposes that the original cells are fibroblasts [28,30]. Other molecular markers of fibroblast heterogeneity were not explored [46]. Nevertheless, these findings indicate that our cells were chiefly fibroblasts.

The accumulated collagen was the main outcome in this study. No significant differences in the capacity to form insoluble collagen by fibroblasts isolated from the rectum or colon were found, which corroborates earlier in vivo measurements [35]. This conclusion is based on the evaluation of 38 different fibroblast strains from 19 colorectal patients. On the other hand, the observed reduction in mRNA levels of *COL1A1* and *COL3A1*, as indicators of reduced type I collagen and type III collagen synthesis in the rectum versus colon, were not translated into fibroblasts in the present study [35]. One possible explanation is that mRNA levels in vivo reflect quiescent conditions whereas culturing activates fibroblasts including their collagen synthesis machinery similar to a wound healing situation.

We estimated collagen deposition using the dye Sirius red to be approximately 20 µg in control cultures. Because Sirius red also binds, albeit weakly, to noncollagenous proteins and stronger to type I than type III collagen, this is most likely an overestimation of the collagen deposited by the fibroblasts in our culture system [47].

AL rates are higher in male versus female colorectal patients, possibly due to reduced collagen biosynthesis [33]. We did not observe significant differences in the collagen parameters examined here between CoReFs derived from males and females, but our sample size was small.

Interestingly, rectal fibroblasts appeared to synthesize more type VI collagen than colonic fibroblasts. Although type VI collagen is important for the architecture of the ECM, the effect of type VI collagen on cell motility is striking. The rate of migration of dermal fibroblasts is increased on fibroblast-generated ECM substrates deficient in type VI collagen [23]. Thus, we hypothesize that type VI collagen impairs fibroblast migration during wound healing. Fibroblast migration was not studied here but would be a valuable addition in refining our cellular model of anastomotic wound repair [48].

TGF-β1 is central in wound healing and fibrosis. TGF-β1 regulates collagen synthesis at several levels [25,27,29]. In support of this hypothesis, TGF-β1 increased the deposition of collagen by CoReFs. TGF-β1 treatment also increased the number of CoReFs after the 8-day treatment period despite attenuating their DNA synthesis after 48 h. Short-term treatment of colonic CRL-1459 fibroblasts with TGF-β1 also reduced DNA synthesis [27]. These seemingly contradictory findings might be explained by the increased formation of sheets with stratified layers of cells and ECM (fibroplasia) with increased duration of incubation, as the effect of TGF-β1 on DNA synthesis and cell proliferation appears to depend on ECM production [49]. Fibronectin matrix, but not collagen, has been suggested to mediate the growth-stimulating effect of TGF-β1 on fibroblasts [49]. TGF-β1 promotes fibronectin synthesis in fibroblasts [48].

Specifically, TGF-β1 increased the synthesis of type I collagen measured by *COL1A1* mRNA and the biomarker PINP but not type III collagen synthesis measured by *COL3A1* mRNA or the biomarker PRO-C3. Treatment of normal human dermal fibroblasts with TGF-β1 also resulted in increased type I collagen but not type III secretion or deposition [50]. Although TGF-β1 increased PRO-C6 in the conditioned media, this was most likely due to concurrent increases in cell numbers, as TGF-β1 reduced *COL6A1* mRNA levels in the CoReFs. Juhl et al. [50] found that TGF-β1 downregulated *COL6A1* in normal human dermal fibroblasts.

In another study, TGF-β did not increase relative collagen synthesis by colonic fibroblasts and reduced the relative collagen synthesis when cultured in the presence of 10% serum (corresponding to approximately 4 ng/mL latent, inactive TGF-β1) [51,52]. These results were from studies on a single cell line (CRL-1459 or CCD-18Co) originating from a 2.5-month-old female.

MMP-2 degrades type I collagen and several other ECM components, and is induced during wound healing [53]. The role of TGF-β1 in the regulation of MMP-2 is complex [54]. In normal human dermal fibroblasts, TGF-β1 seems to increase *MMP-2* mRNA and protein levels [55,56]. The response of CoReFs was the opposite, and TGF-β1 reduced both *MMP-2* mRNA and protein levels as well as MMP activity measured by the C3M biomarker on a cell basis.

In an animal model, the TGF-β1 gene was administered locally to the colonic anastomosis via a viral vector. The application of TGF-β1 on postoperative Day 3, but not on Day 0, resulted in improved anastomotic wound healing on postoperative Day 6 [34]. Thus, timing seems crucial to achieve positive effects of TGF-β1. Brenmoehl et al. [48] found that fibroblast migration depended on the duration of TGF-β1 exposure; 2-day exposure increased migration while exposure for 6 days decreased fibroblast migration. It is possible that the early downregulation of type VI collagen found here plays a role in this altered cellular behavior.

The microbiome of the gastrointestinal tract influences the outcome of colorectal surgery and AL has been associated with specific microorganisms [57,58,59]. The factors responsible for AL have not been identified.

Collagen formation during anastomotic wound healing is important not only to reestablish the collagen network but also to contribute to the restitution of the epithelial layer [60].

In conclusion, we found no effect of anatomic localization on collagen production by fibroblasts derived from the large bowel. TGF-β1 represents a potential therapeutic agent for the prevention of AL by increasing collagen synthesis and collagen deposition. It would be extremely valuable to systematically examine the impact of the secretome of the microorganisms causing AL in our cellular model. The usefulness of the biomarkers examined here to predict AL and monitor the effect of novel treatments would also be worthy of exploration.

## 4. Materials and Methods

### 4.1. Ethical Statements

This study was approved by The Committee on Biomedical Research Ethics for the Capital Region of Denmark (H-17011833), registered by The Danish Data Protection Agency (BFH-2017-058), and conducted at the Digestive Disease Center, Bispebjerg Hospital, University of Copenhagen, Copenhagen, Denmark.

### 4.2. Patients

Patients older than 18 years with colorectal cancer who underwent elective laparoscopic sigmoid resection (SR) or low anterior resection (LAR) with primary anastomosis using a circular stapler were invited to participate [35]. Patients subjected to emergency operation, inflammatory bowel disease, active diverticulitis, ileorectal or ileocolic anastomosis, or interpreter care were excluded. Patients who received glucocorticoids within 2 weeks before the operation were also excluded. Patients were enrolled after providing written informed consent. Demographics and anastomosis localization are shown in Table 4.

### 4.3. Chemicals and Reagents

Chemicals and reagents were purchased from Sigma-Aldrich (St. Louis, MO, USA) unless stated otherwise.

### 4.4. Isolation of Primary Fibroblasts, Maintenance, and Flow Cytometry

Proximal and distal tissue rings were collected during the laparoscopic surgical procedure [35]. Macroscopically healthy tissue rings were washed in ice-cold sterile Hank’s balanced salt solution containing 1.25 mM Ca^2+^, 0.80 Mg^2+^, 5.6 mM glucose, and antibiotics-antimycotics (100 U/mL penicillin, 100 μg/mL streptomycin, and 0.25 μg/mL amphotericin B). The mucosa was scraped off with a knife, including the lamina propria with subepithelial myofibroblasts [62], and the external muscularis externa was macrodissected away. The remaining tissue representing primarily the submucosa was cut into small pieces (1–4 mm). Tissue explants were placed on a tissue culture dish (Cellstar^®^, Greiner Bio-One, Frickenhausen, Germany), air-dried for 20 min, and incubated in 5 mL of complete culture medium (Dulbecco’s modified Eagle medium with high glucose [25 mM], sodium pyruvate, and GlutaMAX™ [DMEM, Gibco] supplemented with 10% fetal bovine serum [FBS, F7524] and the antibiotic-antimycotic mixture) in a humidified atmosphere of 5% CO_2_/air at 37 °C, as shown in Figure 9A. Medium was changed twice weekly. We found that outgrowth over the first 6 days was independent of explant size in the range from 2 to 4 mm, as shown in Figure 9B. After 25–29 days, the outgrown cells (no epithelium was observed) were detached by trypsinization and incubated in 5 mL complete culture medium in T25 tissue culture flasks (Nunc, Roskilde, Denmark).

#### 4.4.1. Subcultivation and Cryopreservation Protocols

Cells were washed in PBS and incubated with 1 mL 0.05% trypsin-0.02% EDTA (Biological Industries, Kibbutz Beit-Haemek, Israel) for 5–6 min at 37 °C in a humidified atmosphere of 5% CO_2_/air. Detached cells were resuspended in 5 mL complete culture medium, and the cell count was performed using a Bürker-Türk hemocytometer or Countess™. Approximately 0.5 × 10^5^ cells were transferred to a new T75 flask. Cells at passage 3 were used for the assays.

Freshly trypsinized cell strains (0.5 × 10^6^ cells) in complete culture medium were centrifuged, and the pellet was suspended in 1 mL cryopreservation medium (70% DMEM, 20% FBS, and 10% dimethyl sulfoxide [Hybri-Max^®^, D2650]) and frozen in a Mr. Frosty™ container (Thermo Fisher Scientific) at −80 °C. Cryopreserved cells were kept at −140 °C for long-term storage.

#### 4.4.2. Flow Cytometry

Cells were grown in a T75-flask to 80% confluence, trypsinized, suspended in PBS, and counted. Cells (3 × 10^5^) were incubated with a (1) FITC-conjugated mouse monoclonal anti-human CD90 (Thy-1) antibody (AS02 clone; Dianova, Hamburg, Germany) diluted 1:50 in PBS [44,45], (2) FITC-conjugated mouse IgG1 isotype control (GM4992; Thermo Fisher Scientific) diluted 1:5 in PBS, or (3) PBS alone for 45 min at 4 °C in the dark. Cells were washed twice with PBS (300× *g* for 5 min), resuspended in PBS, and added to microtiter plates for flow cytometry analysis (Cell Lab Quanta SC MPL, Beckman Coulter, Brea, CA, USA). Data were analyzed using Kaluza software (Beckman Coulter).

### 4.5. Experimental Design and Execution of the Assay for Collagen Deposition and Cell Density

We adopted the method of Trackman et al. [29] to measure collagen accumulation with Sirius red staining and cell density with crystal violet staining. Cells were seeded (6 × 10^4^ cells/well) in 6-well tissue culture plates (Nunclon™ Delta Surface, Nunc, Roskilde, Denmark). Cells were incubated to 70–80% confluence in complete culture medium (3 mL/well), which took a mean of 5 days (range: 3–7 days) in culture. The medium was then removed and replaced with assay culture medium (complete culture medium with reduced FBS concentration [1% FBS] supplemented with 50 μg/mL L-ascorbic acid [A4544, Sigma-Aldrich]). Cells were then incubated for a total of 8 days with collection of conditioned media with replacement with fresh assay culture medium after 4 days. After 8 days, conditioned media were again collected and stored at −80 °C until analysis. Cells from the last 12 analyzed patients were treated with or without 1 ng/mL rhTGF-β1 (240-B; R&D Systems, Minneapolis, MN, USA). rhTGF-β1 reconstituted in 1 mg bovine serum albumin (A3803)/mL 4 mM HCl was added to 2 separate wells (30 µL of 100 ng/mL TGF-β1/well), and the remaining 4 wells received control alone (30 µL of 1 mg BSA/mL 4 mM HCl/well). The dose-response effect of TGF-β1 was studied in 6 of 12 patients. Fibroblasts in two 6-well tissue culture plates from each patient were used; in each plate, 3 wells received control, while the other 3 wells received 0.01, 0.1, or 1 ng/mL TGF-β1. To estimate the amount of collagen deposited by the fibroblasts, neutralized type I collagen (0.5 mg/mL) from rat tails (354236; Corning, Bedford, MA, USA) was dispensed into an empty 6-well plate, incubated in a humidified atmosphere of 5% CO_2_/air at 37 °C for 2 h and dried in a laminar flow bench. The plate was then processed identically to the plates with cultured cells. The wells were washed in PBS, fixed in 2.5 mL Bouin’s solution for 1 h, washed in running tap water, and air-dried overnight.

The fixed cells/collagen were stained with 2 mL Direct Red 80 dye (Fluka 43665) at 1 mg/mL water-saturated picric acid (1.3%) for 1 h on an orbital shaker at 100 rpm. The nonbound dye was removed, and the wells were washed 4 times with 0.01 M HCl. The bound dye was eluted in 1.0 mL 0.1 M NaOH for 30 min on the orbital shaker. The optical density (OD_540nm_) was measured in 200 µL duplicate aliquots in a microtiter plate against the blank 0.1 M NaOH. Cells were then washed with PBS and stained with 3 mL crystal violet (Merck 1.01408) in deionized water at 1 mg/mL for 30 min on an orbital shaker. The superfluous dye was washed away in running tap water, and the stained cells were air-dried overnight. The stain was eluted with methanol (5 mL/well) and the OD_590nm_ measured in 200 µL duplicate aliquots with methanol as blank in a 96-well microtiter plate. The mean OD_540nm_ and OD_590nm_ measurements of the 200 µL aliquots are reported.

### 4.6. Collagen Biomarker Assays

PINP [36], PRO-C3 [37], PRO-C6 [38], and C3M [39] levels were measured in conditioned media by competitive ELISAs at Nordic Bioscience (Herlev, Denmark) according to the manufacturer’s instructions. The sum of released biomarkers at Days 4 and 8 was used for statistical analyses.

### 4.7. mRNA Analyses by RT-qPCR

The effect of TGF-β1 on *COL1A1*, *COL3A1*, *COL6A1*, *MMP2*, and *ACTA2* mRNA levels was studied in CoReFs (5 × 10^4^ in 1 mL/well) grown for 6 days to confluence in 12-well tissue culture plates (Cellstar^®^, Greiner Bio-One) in complete culture medium. The medium was removed and the CoReFs were treated for 48 h with or without 1 ng/mL TGF-β1 in assay culture medium. Total RNA was extracted from the treated cells with 1 mL TriReagent^®^ (Molecular Research Center, Cincinnati, OH, USA). Bromochloropropane (100 µL) was added to isolate the aqueous phase containing the RNA, which was precipitated using isopropanol. The RNA pellet was then washed in ethanol and subsequently dissolved in 10 μL RNAse-free water. Total RNA concentrations were determined with the RiboGreen assay (R11490; Thermo Fisher Scientific). Total RNA (500 ng) was converted into cDNA in 20 μL using OmniScript reverse transcriptase (Qiagen, Valencia, CA, USA) and 1 μM poly-dT (Thermo Fisher Scientific) according to the manufacturer’s protocol (Qiagen). For each target mRNA, 0.5 μL cDNA was amplified in 25 μL SYBR Green polymerase chain reaction (PCR) containing 1× Quantitect SYBR Green Master Mix (Qiagen, Hilden, Germany) and 100 nM of each primer, as shown in Table 5. The amplification was monitored in real time using an MX3005P Real-time PCR machine (Stratagene, La Jolla, CA, USA). Ct values were related to a standard curve made with known concentrations of cloned PCR products or DNA oligonucleotides (Ultramer™ oligos, Integrated DNA Technologies, Leuven, Belgium) with a DNA sequence corresponding to the sequence of the expected PCR product. The specificity of the PCR products was confirmed by melting curve analysis after amplification. *GAPDH* mRNA was chosen as the internal control.

### 4.8. WST-1 and BrdU Assays

Cells were seeded into 96-well plates (TPP^®^, Trasadingen, Switzerland) at 1 × 10^4^ cells per well in complete culture medium and incubated for 4 days. The medium was then replaced with 200 μL assay culture medium/well with or without 1 ng/mL TGF-β1, and cells were incubated for another 48 h.

WST-1 reagent (05015944001; Roche Diagnostics, Mannheim, Germany) was added (20 µL/well) 0.5 h before the end of the 48 h treatment period, and ΔOD (OD_450nm_-OD_630nm_) was measured on a microplate reader (800TS; BioTek Instruments, Winooski, VT, USA).

BrdU in DMEM with 1% FBS was added to all wells (20 μL per well) to a final concentration of 10 μM, except for the background control, to which 20 μL of DMEM with 1% FBS alone was added, and was present during the final 4 h of incubation. The cells were then washed three times with PBS, fixed, and incubated with a mouse monoclonal anti-BrdU antibody conjugated with peroxidase (1:50 dilution) according to the manufacturer’s instructions (11647229001; Roche Diagnostics). The immune complex was detected by the tetramethyl-benzidine substrate (TMB ONE; Kem-En-Tec Diagnostics, Taastrup, Denmark), and the reaction stopped after 20 min with 0.2 M H_2_SO_4_ and the ΔOD (OD_450nm_–OD_630nm_) was measured on a microplate reader.

### 4.9. Immunofluorescence for α-SMA

Immunofluorescence was performed in Nunc™ Lab-Tek™ 4-well (1.8 cm^2^/well) chamber glass slides essentially following the procedures described by Smith-Clerc and Hinz [63]. The wells were incubated with DMEM plus 10% FBS at 37 °C and 5% CO_2_ for 1 h before CoReFs (1 × 10^4^ cells/well) in 0.5 mL of DMEM with 2% or 10% FBS were seeded. After one day, TGF-β1 (1 ng/mL) was added, and the cells were incubated for 4 days. The media were removed, and the cells were washed 3 times in PBS and then fixed in 3% paraformaldehyde/PBS (1 mL/well) for 10 min at room temperature (RT). Fixed cells were permeabilized with 0.2% Triton X-100 (TX-100) in PBS for 5 min at RT (2 mL/well) and incubated with a mouse monoclonal antibody against human α-SMA (1A4 clone, M085129; Agilent Technologies, Glostrup, Denmark) diluted in PBS/0.02% TX-100 to 2 µg/mL (250 μL/slide) overnight at 4 °C [48]. Cells were washed with PBS/0.02% TX-100 and then incubated with Alexa Fluor^®^ 488-conjugated goat anti-mouse IgG (A11001; Thermo Fisher Scientific) secondary antibody diluted in PBS/0.02% TX-100 to 1 µg/mL (250 µL/slide). Cells were washed with PBS/0.02% TX-100 and deionized water at RT and then incubated with Hoechst 33342 (H1399; Thermo Fisher Scientific) at 2.5 µg/mL at RT for 5 min in the dark (250 μL/slide). After PBS washes, cells were cover-slipped with Fluorescence Mounting Medium (S302380; Agilent Technologies).

### 4.10. MMP-2 in Conditioned Media Analyzed by Gelatin Zymography

Gelatin zymography was performed with 15 wells using a 1.0 mm thick 10% Tris–glycine SDS-PAGE gel containing 1 mg/mL gelatin. Conditioned medium samples (5 µL) were mixed with 5 μL of 2× Laemmli sample buffer (#1610737; Bio-Rad Laboratories, Hercules, CA, USA) and electrophoresed at a constant voltage of 125 V for 105 min. The rhMMP-2 (PF037; Calbiochem^®^) standard was run in parallel with the samples. The gelatin-copolymerized gel was renatured in 25 μL/mL TX-100 for 30 min at ambient temperature and incubated at 37 °C for 18 h in developing buffer composed of 50 mM Tris-HCl (pH 7.5), 10 mM CaCl_2_, 1 μM ZnCl_2_, 1 μL/mL TX-100, and 0.2 mg/mL NaN_3_. The gel was stained with the Colloidal Blue Staining Kit (Thermo Fisher Scientific), destained, mounted, and scanned. The amounts of MMP-2 were estimated by densitometry using ImageJ software (1.52a; National Institutes of Health, Bethesda, MD, USA) and presented as pg [64].

### 4.11. Statistical Analyses

The effect of localization, sex, and TGF-β1 on collagen deposition and cell density was analyzed by a paired or unpaired Student’s *t*-test, and biomarker levels were analyzed by the Wilcoxon signed-rank or Mann–Whitney U tests. For analyses of the effect of sex and TGF-β1, collagen deposition, cell density, and biomarker results were averaged from fibroblasts isolated from the proximal and distal tissue rings. The effect of TGF-β1 on log-transformed mRNA levels was analyzed by a paired Student’s *t*-test. The TGF-β1 dose-response experiments were analyzed with one-way ANOVA with repeated measures. Two-sided statistical analyses were performed using SPSS Statistics 26.0 software (IBM, Armonk, NY, USA). The statistical significance level was defined as *p* < 0.05. Data are presented as the mean ± SEM or median (interquartile range).

## Figures and Tables

**Figure 1 ijms-22-01616-f001:**
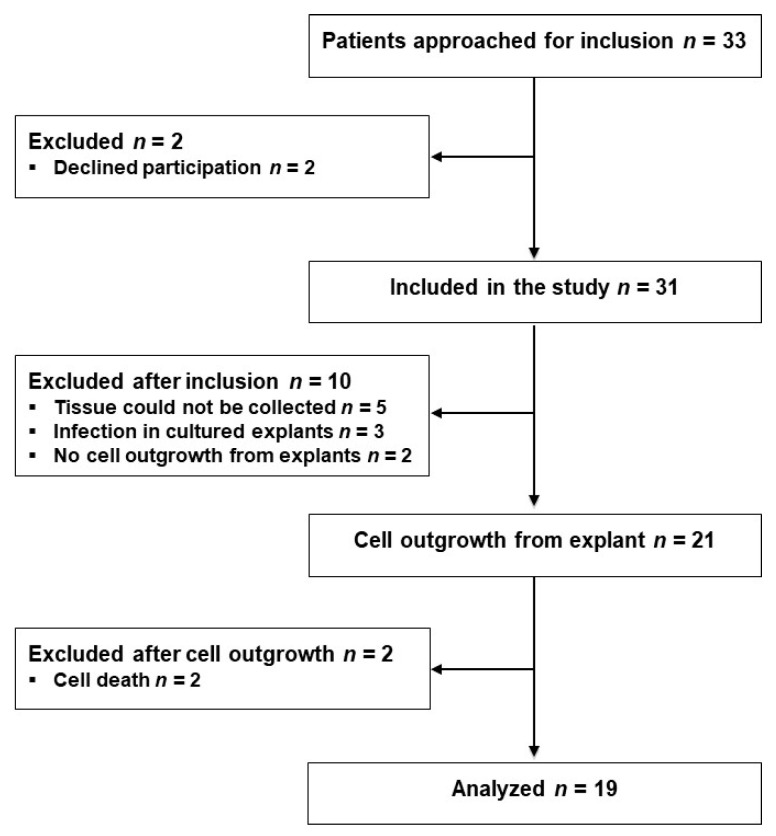
Flow diagram of the study participants.

**Figure 2 ijms-22-01616-f002:**
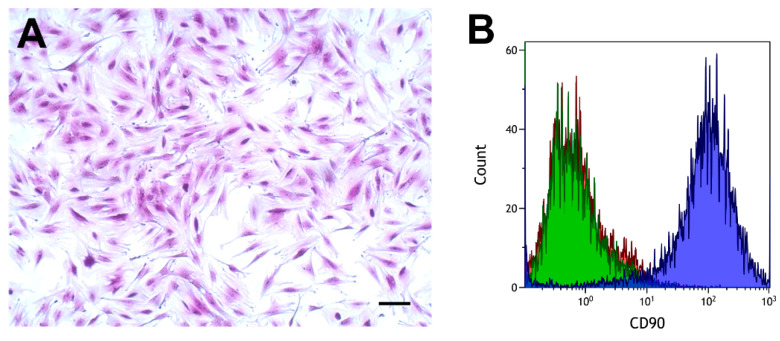
Morphology (**A**) and flow cytometry analysis of colorectal fibroblasts (**B**). (**A**) Crystal violet stain. Scale bar, 100 µm. (**B**) CD90, blue curve; IgG1 isotype control, green curve; phosphate-buffered saline (PBS), red curve.

**Figure 3 ijms-22-01616-f003:**
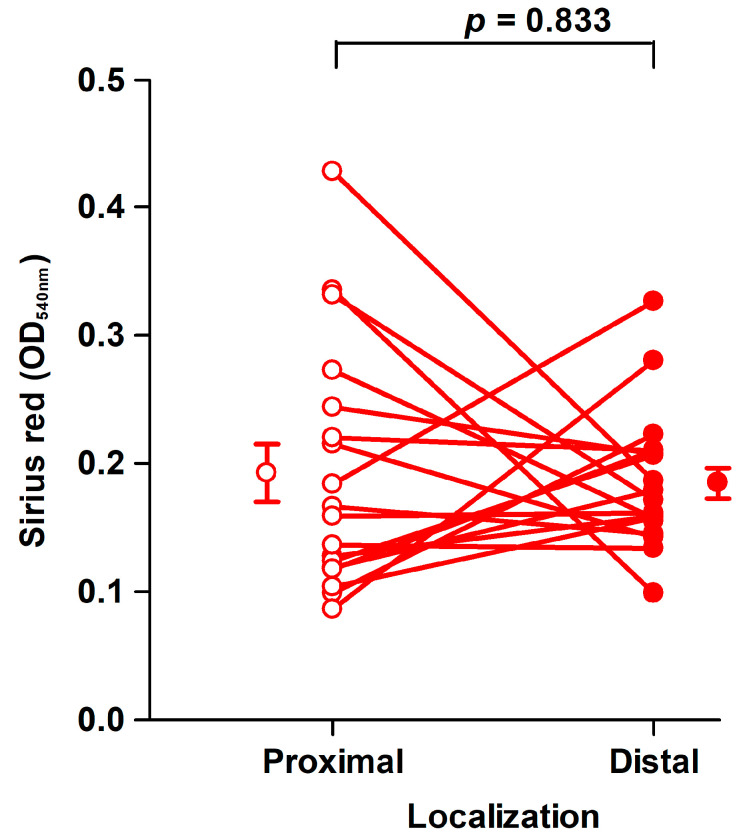
Collagen deposition by fibroblast strains cultured for 8 days. There was no significant difference in the amount of Sirius red eluted from fibroblasts derived from proximal (open circles, OD_540nm_: 0.19 ± 0.023/well) and distal (filled circles, OD_540nm_: 0.18 ± 0.012/well) tissue. Mean ± SEM (standard error of the mean) (*n* = 19). An OD_540n__m_ value of 0.20 corresponds to 20 µg type I collagen.

**Figure 4 ijms-22-01616-f004:**
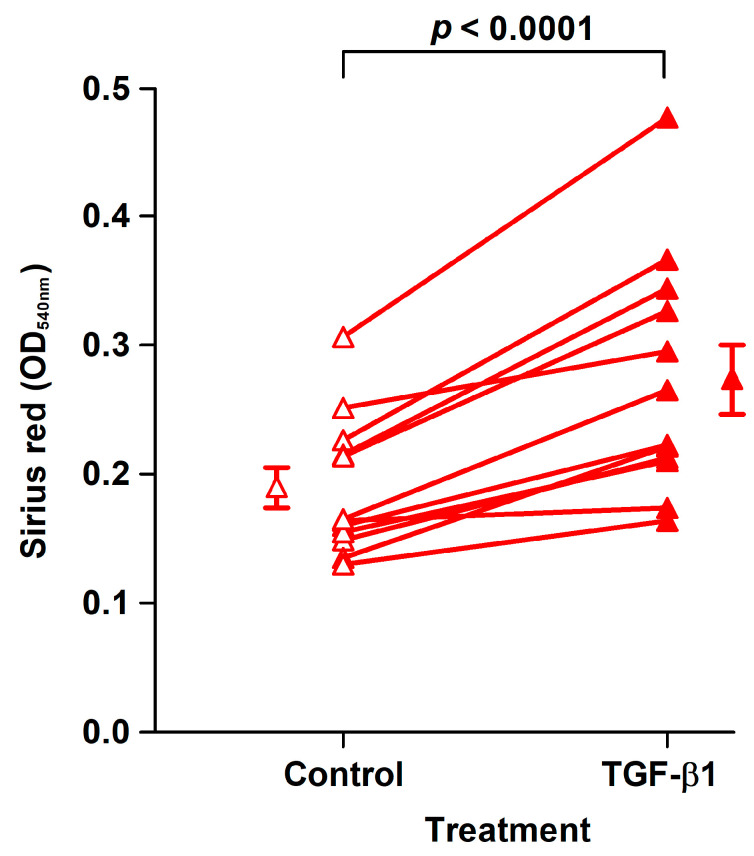
TGF-β1 treatment (closed triangles, OD_540nm_: 0.27 ± 0.027/well) of CoReFs for 8 days increased collagen deposition compared to the control (open triangles, OD_540nm_: 0.19 ± 0.015/well), as measured by Sirius red staining. An OD_540nm_ value of 0.20 corresponds to 20 µg type I collagen. Mean ± SEM (*n* = 12).

**Figure 5 ijms-22-01616-f005:**
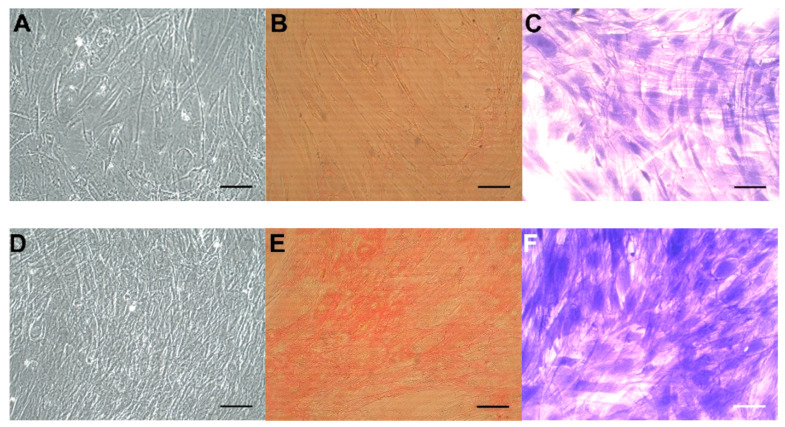
CoReFs treated without (**A**–**C**) or with 1 ng/mL TGF-β1 for 8 days (**D**–**F**), fixed and stained with Sirius red (**B**,**E**) or crystal violet (**C**,**F**). OD_540nm_ for (**B**) was 0.22, and for (**E**) was 0.59. OD_590nm_ for (**C**) was 0.54 and for (**F**) was 1.46. Scale bars, 100 µm. Cells in (**A**) and (**D**) were trypsinized (2 mL 0.05% trypsin-0.02% EDTA/well for 10 min at 37 °C) and counted by a Countess™ automated cell counter (Thermo Fisher Scientific, Waltham, MA, USA). The number of cells in (**A**) was 0.77 × 10^6^ and in (**D**) was 1.2 × 10^6^.

**Figure 6 ijms-22-01616-f006:**
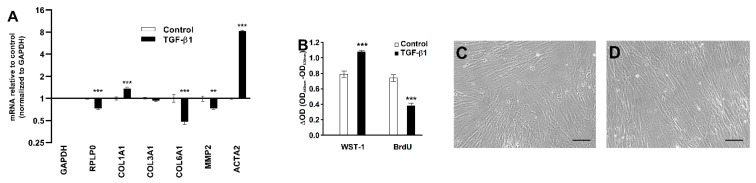
Effect of TGF-β1 (1 ng/mL) treatment of CoReFs for 48 h on *COL1A1*, *COL3A1*, *COL6A1*, *MMP2*, and *ACTA2* mRNA levels (**A**) and cell proliferation measured by WST-1/BrdU assays (**B**). Phase contrast photomicrographs (Olympus CK40 with Moticam^®^ S2) of CoReFs treated without (**C**) or with TGF-β1 (**D**) for 48 h just before adding the WST-1 reagent. Scale bars, 100 µm. (**A**) mRNA levels were normalized to *GAPDH*, expressed as fold change relative to control-treated CoReFs and log-transformed. Geometric mean ± back-transformed SEM of 6 replicates is shown. (**B**) Mean ± SEM of 10 replicates is shown. ** *p* < 0.01, *** *p* < 0.001 vs. control.

**Figure 7 ijms-22-01616-f007:**
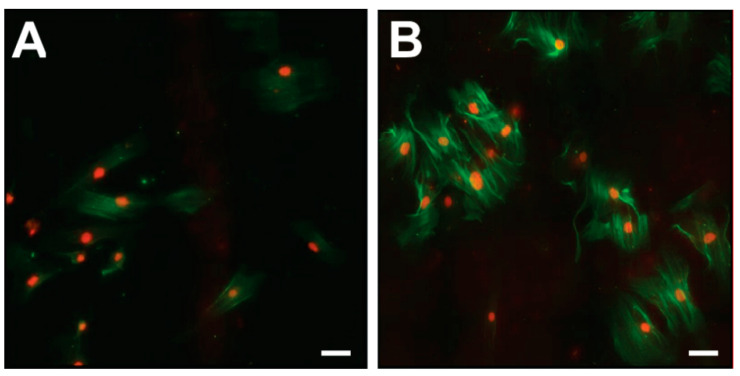
Immunofluorescence for α-SMA (green) in CoReFs treated without (**A**) or with 1 ng/mL TGF-β1 (**B**) for 4 days. Scale bars, 20 µm.

**Figure 8 ijms-22-01616-f008:**
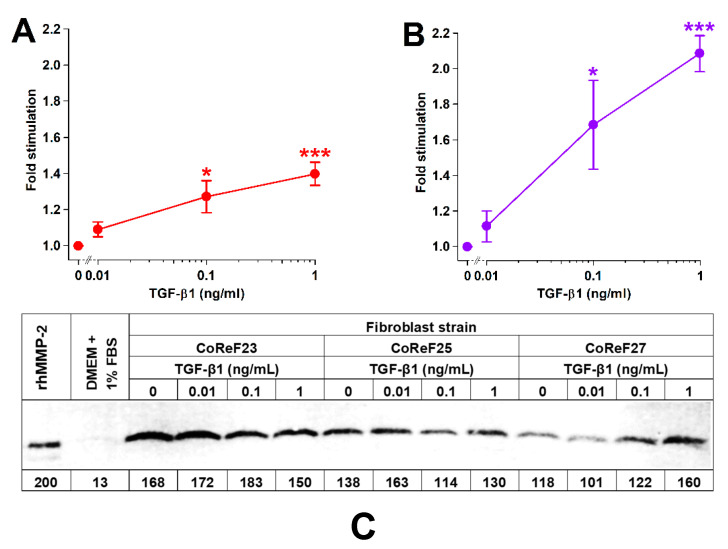
Effect of TGF-β1 on collagen deposition (**A**), cell numbers (**B**), and MMP-2 (**C**) of CoReFs treated for 8 days at three different concentrations (0 [control], 0.01, 0.1 and 1 ng/mL). (**A**,**B**) Mean ± SEM (*n* = 6). * *p* < 0.05, *** *p* < 0.001 vs. control. (**C**) A representative zymogram (inverse image) of MMP-2 in the conditioned media (5 µL/lane) from CoReFs (3 patients). The estimated MMP-2 content (pg) is indicated below each lane of the zymogram from the densitometric determination of band intensities by ImageJ.

**Figure 9 ijms-22-01616-f009:**
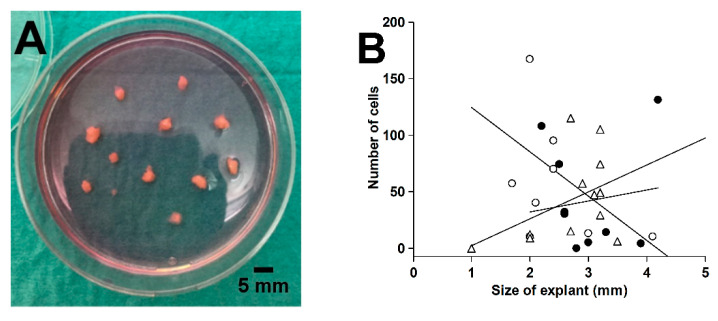
Explanted colorectal tissue for fibroblast expansion (**A**) and the effect of explant size on the cell outgrowth from 3 different tissue rings after 6 days in culture (**B**). Linear regression lines for the 3 tissues are shown in (**B**).

**Table 1 ijms-22-01616-t001:** Effect of localization on the cumulative release of PINP, PRO-C3, PRO-C6, and C3M biomarkers from fibroblasts into the media over 8 days of culture shown as ng/well ^1^.

Collagen Biomarker	Proximal	Distal	*p*-Value ^2^
PINP	1580 (855–2290)	2620 (655–4190)	0.0854
PRO-C3	3.7 (2.0–4.9)	3.7 (1.5–4.3)	0.184
PRO-C6	4.2 (2.3–6.2)	6.7 (3.2–14.2)	0.00497
C3M	2.0 (1.7–2.8)	2.9 (1.4–3.9)	0.125

^1^ Median (interquartile range), *n* = 19. ^2^ Wilcoxon signed-rank test.

**Table 2 ijms-22-01616-t002:** Effect of sex on cumulative release of PINP, PRO-C3, PRO-C6, and C3M biomarkers by colorectal fibroblasts (CoReFs) into the media over 8 days of culture shown as ng/well. ^1.^

Collagen Biomarker	Males	Females	*p*-Value ^2^
PINP	1470 (716–3460)	2650 (2090–3620)	0.206
PRO-C3	4.0 (2.8–4.9)	3.0 (1.8–4.4)	0.206
PRO-C6	5.0 (4.1–6.0)	6.7 (2.4–11.4)	0.778
C3M	2.2 (1.6–2.6)	2.9 (2.3–3.5)	0.174

^1^ Median (interquartile range). ^2^ Mann–Whitney U test.

**Table 3 ijms-22-01616-t003:** Effect of TGF-β1 treatment of the CoReFs on cumulative release of PINP, PRO-C3, PRO-C6, and C3M biomarkers into the media over 8 days of culture shown as ng/well ^1^.

Collagen Biomarker	Control	TGF-β1	*p*-Value ^2^
PINP	2520 (2040–3340)	4550 (3310–6660)	0.00222
PRO-C3	3.4 (2.6–4.6)	3.9 (2.2–4.8)	0.666
PRO-C6	5.4 (3.8–10.3)	9.3 (6.6–12.1)	0.00221
C3M	2.3 (1.7–2.9)	2.3 (1.6–2.8)	0.593

^1^ Median (interquartile range), *n* = 12. ^2^ Wilcoxon signed-rank test.

**Table 4 ijms-22-01616-t004:** Patient characteristics ^1^.

	Included	Analyzed
Number of patients	31	19
Males/females	18/13	11/8
Age (years)	64.5 ± 2.5	62.6 ± 2.4
Body mass index (kg/m^2^)	26.3 ± 1.0	26.7 ± 1.4
Diabetes mellitus	5	4
Smoker	7	4
ASA physical status classification of 2014 [61] ^2^		
I	13	7
II	15	9
III	3	3
Type of resection ^3^		
SR	20	13
LAR	11	6

^1^ Values are the mean ± SEM or represent numbers of patients. ^2^ ASA, American Society of Anesthesiologists. ^3^ SR, elective laparoscopic sigmoid resection; LAR, elective laparoscopic low anterior resection.

**Table 5 ijms-22-01616-t005:** Primer sequences for RT-qPCR.

mRNA	GenBank^®^ ID	Sense (Forward)	Antisense (Reverse)
*COL1A1*	NM_000088.3	GGCAACAGCCGCTTCACCTAC	GCGGGAGGTCTTGGTGGTTTT
*COL3A1*	NM_000090.3	CACGGAAACACTGGTGGACAGATT	ATGCCAGCTGCACATCAAGGAC
*COL6A1*	NM_001848.2	CACACCGCTCAACGTGCTCTG	GCTGGTCTGAGCCTGGGATGAA
*MMP2*	NM_004530.5	CCGCCTTTAACTGGAGCAAAAACA	TTGGGGAAGCCAGGATCCATTT
*ACTA2*	NM_001613.4	AGAAGGAGATCACGGCCCTAGCA	CCCGGCTTCATCGTATTCCTGT
*GAPDH*	NM_002046.4	CCTCCTGCACCACCAACTGCTT	GAGGGGCCATCCACAGTCTTCT
*RPLP0*	NM_053275.3	GGAAACTCTGCATTCTCGCTTCCT	CCAGGACTCGTTTGTACCCGTTG

## Data Availability

The data presented in this study are available on request from the corresponding author.

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
