# Peer review of "A Human Cellular Model for Colorectal Anastomotic Repair: The Effect of Localization and Transforming Growth Factor-β1 Treatment on Collagen Deposition and Biomarkers"

_ijms, 2021, doi:10.3390/ijms22041616_

Round 1
Reviewer 1 Report
The authors in their study highlighted how TGF-β1 can help in the prevention of Anastomotic leakage.
The manuscript is very interesting.
Could the authors explain whether they found a difference between the males and females?
The authors state that the microbiota is also interested, why haven't they evaluated it?
I suggest that the authors read these recent articles which perhaps could further improve the manuscript. Chao G, Wang Z, Chen X, Zhang S. Cytokines in the colon, central nervous
system and serum of irritable bowel syndrome rats. Eur J Med Res. 2021 Jan 13; 26 (1): 7.
Panahipour L, Omerbasic A, Nasirzade J, Gruber R. TGF-β Activity of a
Demineralized Bone Matrix. Int J Mol Sci. 2021 Jan 11; 22 (2): E664.
Do the authors plan to continue the studies by increasing the number of patients?
Author Response
Thank you for the very useful comments.
- We have now included data from males and females (new section and Table 2) on lines 135-151 and discussed these results (lines 260-263). The analyses showed no statistically significant differences between males and females. Our study was not designed to study this question and would require inclusion of more patients to be more definitive about our conclusion.
- We have not studied the effect of the microbiota in our cellular model. These are extensive studies and would require substantial efforts, which we feel goes beyond the scope of the present manuscript.
- Thank you for highlighting these two interesting articles. However, we could not find applications of the aims, methods or results to our specific aims of our project.
- At the moment we do not plan to include more patients but use our biobank of the cryopreserved colorectal fibroblasts.
Reviewer 2 Report
Anastomotic leakage (AL) is a devastating complication after colorectal surgery possibly 19 due to loss of stabilizing collagen fibers in the submucosa.
The authors assess the formation of collagen in colon versus rectum with or without transforming growth factor (TGF)-β1 exposure in a 21 human cellular model of colorectal repair.
From the results obtained they concluded that TGF-β1 is a potential therapeutic agent for prevention of AL by increasing type I collagen synthesis 35 and collagen deposition.
My suggestions is to add scale bar at the pictures of fig. 2A and 7 and write the appropriate legends as indicated in other figures.
Author Response
Thank you for the suggestion. We have now included scale bars in Figure 2A and Figure 7.